# Economic Impacts of Horticulture Research and Extension at MSU Coastal Research and Extension Center

Benedict C. Posadas [1],*, Patricia R. Knight [1], Eric T. Stafne [1], Christine E. H. Coker [1], Gary Bachman [1], James DelPrince [1], Scott A. Langlois [1] and Eugene K. Blythe [2]

[1] Coastal Research and Extension Center, Mississippi Agricultural and Forestry Experiment Station, Mississippi State University, Biloxi, MS 39532, USA; patricia.knight@msstate.edu (P.R.K.); eric.stafne@msstate.edu (E.T.S.); cec117@msstate.edu (C.E.H.C.); gary.bachman@msstate.edu (G.B.); jdelprince@pss.msstate.edu (J.D.); scott.langlois@msstate.edu (S.A.L.)
[2] College of Agriculture, Auburn University, Auburn, AL 36849, USA; blythek@auburn.edu
* Correspondence: ben.posadas@msstate.edu

**Abstract:** This paper summarizes the estimates of the total changes in sales, expenses, and income of participants of the horticulture research and extension programs at the Mississippi State University—Coastal Research and Extension Center for the past five years. Major items outline the estimation procedures for the past five years. The average annual values were used in estimating the total economic impacts of added gross sales, expenses, and incomes of participants in horticulture events. The cumulative total impacts reach USD 8.7 million in sales, 76 jobs, USD 1.4 million in labor income, USD 2.4 million in value-added, and USD 0.4 million in local, state, and federal taxes. In addition, the total willingness to pay for the horticulture program by the adult participants reached USD 1.8 million. In comparison, the annual public spending on the horticulture program averaged USD 1.4 million, creating additional substantial economic impacts to the region.

**Keywords:** economic impact; ornamental horticulture; fruits and nuts; vegetables; flowers; horticulture research; extension





## 1. Introduction

Horticulture research and extension are essentially public goods funded for the benefit of the public [1]. Economic impact assessments are necessary to justify the continued funding of these research and extension programs. As agricultural research budgets are being subjected to strict scrutiny, research centers or programs need to show that they are worth the investment of state, federal, or industry funds [2]. This is especially so under "significant downsizing of public support for agricultural research and development (R&D) and a major decline in the share of that research devoted to preserving or promoting productivity growth" [3]. Budgetary pressures in recent years have restricted investment in public agricultural science research, extension, and infrastructure [4]. The benefits from past public investments in agricultural research have been worth many times more than the costs [5].

Horticulture researchers and extension specialists rely on industry feedback to document that the research-based information is timely and relevant to their emerging and existing issues [1]. "Agricultural extension services are among the most common forms of public-sector support of knowledge diffusion" [6]. "Extension programs provide technical education services to farmers through demonstrations, lectures, contact with farmers, and other media" [7]. Extension specialists provide farmers with technical information to better evaluate new technology before adoption and communicate feedback to technology suppliers. "Extension plays an important role in disseminating new technology and bridging the gap between innovation in the laboratories and practice on the farm" [8].

The sequence of extension impact can be described as follows [6]: (1) extension information along with information from other sources is shared, (2) knowledge formation leads

to farmer experimentation, (3) gradual adoption of new practice takes place if innovation appears productive, and (4) with the adoption of new technology, changes in input use take place, outputs increase, and production costs are reduced. The shaping of knowledge and observations from producers most likely leads to other producers' adoption of new methods. As specialists convey information to industry professionals, program evaluations document that the specialist expressed relevant data and did it in a readily usable format.

The environmental horticulture or green industry complex includes input suppliers, production firms, wholesale distribution firms, horticultural service firms, and retail operations [9]. The Mississippi green industry has exhibited consistent growth. Sales contributions expanded from less than USD 1 billion in 2002 to almost USD 2.5 billion in 2018 [10–13]. More jobs were created by the entire industry, expanding from approximately 14,000 jobs in 2002 to nearly 17,000 jobs in 2018.

The overall goal of this paper is to present a quantitative assessment of the economic impacts of the horticultural research and extension programs at the Mississippi State University (MSU)—Coastal Research and Extension Center (CREC). The specific objectives of this paper are as follows:

- Develop a systematic methodology in quantifying the total changes in private spending, sales, income, and willingness to pay by participants of horticulture research and extension programs.
- Estimate the total changes in private spending, sales, income, and willingness to pay by participants of the research and extension programs on ornamental horticulture, vegetables, fruits, and nuts during the last five years.
- Calculate the total economic impacts of the total changes in private spending, sales, and income by participants who attended the horticulture events during the past five years.

It is expected that the application of new horticultural information will facilitate households, non-profit organizations, and businesses to increase sales or funding, reduce costs, or increase savings in their homes or organizations [1]. Before adopting or rejecting these new methods, they will evaluate their usefulness to their respective households or businesses. The decisions made by other households or enterprises regarding the adoption or rejection of new methods will also influence their choices. Expected benefits and costs will ultimately determine the adoption or rejection of new horticulture methods by businesses, households, and non-profit organizations.

## 2. Materials and Methods

### 2.1. Sources of Primary Data

The survey estimates of the total changes in spending, sales, and income of participants of the horticulture research and extension programs at the MSU-CREC were conducted at its experiment stations and extension offices located in Coastal Mississippi [1]. In these, attendees were asked to participate in voluntary surveys conducted after some of these events.

Since data were primarily collected from voluntary surveys and self-reporting, large standard deviations from the mean were observed. The earlier analysis incorporated the standard deviations of each economic variable. However, it was decided to drop the variations and use only the mean of the available data to estimate economic impacts.

### 2.2. Number of Participants

Participants consisted of adults who attended horticulture events at the MSU-CREC research and extension facilities [1]. During the past five years (2015–2019), the annual number of adult attendees to horticulture events averaged over 1600 persons or a total of more than 8400 producers, Master Gardeners, and research and extension personnel.

The Muscadine Field Days and Workshops (M-FD-WS) averaged more than 150 producers per year. About 70 producers per year participated in the Blueberry Field

Days and Workshops (BB-FD-WS). The Beaumont Vegetable Field Days (BV-FD) are held annually, with an average attendance of 65 producers.

The MSU-CREC annual Producer Advisory Council meeting (CREC-PAC) was attended by more than 100 horticultural producers and practitioners. Floral Workshops (FLORAL-WS) topped the list with almost 1200 floral enthusiasts per year. The Ornamental Horticulture Field Days (OH-FD) averaged about 100 attendees per year, mainly Master Gardeners and producers.

### 2.3. Changes in Participants' Horticulture Spending, Sales, and Income

The attendees of the OH-FD in 2017 and 2019 were asked to participate in a survey of their opinions about the horticulture research and extension activities at MSU-CREC [1]. The estimates of the average changes in horticulture spending, sales, income, and willingness to pay for the horticulture research and extension programs at MSU-CREC are discussed below.

The floral registration fees consisted of fees collected from participants of the various floral programs conducted by MSU-CREC from 2016 to 2019. Total registration fees averaged $USD 11,632/year from their initial start in 2016 until 2019. Registration fees were collected only from floral workshop attendees.

The participants' annual travel expenses included distance traveled, meals and hotel, airfare and baggage fees, and other expenses. Travel expenses reported by the participants at the OH-FD in 2017 and 2019 averaged $USD 63/person/year. The annual participants' travel costs to the five horticulture programs during the past five years were computed as follows:

$$\text{Travel cost (\$USD/year)} = \text{number of participants} \times [\text{distance travelled (miles/person/year)} \times \text{cost per mile (\$USD/mile)} + \text{meals and hotel expenses (\$USD/person/year)} + \text{airfare and baggage fees (\$USD/person/year)} + \text{other expenses (\$USD/person/year)]} \tag{1}$$

The increase in annual gross sales reported by the participants at the OH-FD in 2017 and 2019 averaged $USD 24/person/year. Project funding increased by an average of $USD 156/person/year. The annual increase in gross sales and project funding for the five horticulture programs from 2015 to 2019 was calculated as follows:

$$\text{Increase in sales and funding (\$USD/year)} = \text{number of participants} \times [\text{gross sales (\$USD/person/year)} + \text{funding increase (\$USD/person/year)]} \tag{2}$$

The annual increase in participants' savings averaged $ USD 146/person/year. The increase in participants' savings for the five horticulture programs was calculated as follows:

$$\text{Increase in savings (\$USD/year)} = \text{number of participants} \times \text{savings increase (\$USD/person/year)} \tag{3}$$

The average annual decrease in participants' costs was $USD 73/person/year. The reduction in participants' costs for the five horticulture programs was calculated as follows:

$$\text{Decrease in costs (\$USD/year)} = \text{number of participants} \times \text{costs decrease (\$USD/person/year)} \tag{4}$$

The willingness to pay (WTP) for the information learned from the horticulture programs conducted by the OH-FD in 2017 and 2019 averaged $USD 1305/person. This average WTP was applied in estimating total values for participants of the OH-FD, CREC-

PAC, and FLORAL-WS. The willingness to pay by OH-FD, CREC-PAC, and FLORAL-WS participants was calculated as follows:

$$\text{Willingness to pay (\$USD)} = \text{number of OH-FD, CREC-PAC, and FLORAL-WS participants} \times \text{willingness to pay at OH-FD (\$USD/person)} \quad (5)$$

A second estimate of the willingness to pay for information learned from BB-FD-WS was $USD 42/person. This secondary estimate was applied to participants of the BB-FD-WS, M-FD-WS, and BV-FD. The willingness to pay by BB-FD-WS, M-FD-WS, and BV-FD participants was estimated as follows:

$$\text{Willingness to pay (\$USD)} = \text{number of BB-FD-WS, M-FD-WS, and BV-FD participants} \times \text{willingness to pay at BB-FD-WS (\$USD/person)} \quad (6)$$

*2.4. Economic Impact Analysis*

Five types of economic impacts were estimated to quantify the annual effects of horticulture research and extension programs: output or sales, employment, income, the total value-added, and tax revenues. Sales, income, total value-added, and tax impacts are expressed in 2019 dollars. Employment impacts are described in terms of a mix of both full-time and part-time jobs. Output or sales are the gross sales by businesses within the state of Mississippi. Labor income includes personal income such as wages and salaries and proprietors' income or income from self-employment. Tax revenues consist of state, local, and federal tax collections.

The total economic impact is the sum of direct, indirect, and induced impacts. Direct impacts are derived from the estimates of the increase in horticulture sales and funding, income increase, and increase in horticulture expenses. Indirect impacts result from changes in the economic activity of other industrial sectors that supply goods or services to the horticulture research and extension sector. Induced impacts are the result of personal consumption expenditures by industry employees.

Total economic impacts of horticulture research and extension programs were estimated by using IMPLAN [14] software. The IMPLAN sector used in the economic impact analysis of research and extension spending was sector 464 (scientific research and development service). For horticulture spending, sales, and income reported by participants, the economic sector was IMPLAN sector 6 (greenhouse, nursery, and floriculture production).

## 3. Results

Most of the participants viewed horticulture research and extension at MSU-CREC as providing helpful information that benefitted their households, businesses, or non-profit organizations [1]. Most OH-FD participants benefitted from new horticulture information learned from MSU-CREC in the last five years [1]. Seventy-four percent of the OH-FD participants gained new knowledge from the MSU-CREC horticulture programs over the previous five years. Among the OHFD attendees, 93% applied new information learned during the last five years to their households, 19% applied it to their research and extension projects, 16% applied it to their non-profit organizations, and 7% applied it to their private businesses [1].

To facilitate the computation of total economic impacts using IMPLAN [1] software and 2019 Mississippi state data, the annual changes in spending, sales, and incomes by participants of the horticulture events conducted by MSU-CREC were summarized (Figure 1). Economic impacts of each horticultural item were individually estimated and compiled to show the total effects in terms of sales, jobs, income, value-added, and tax revenues.

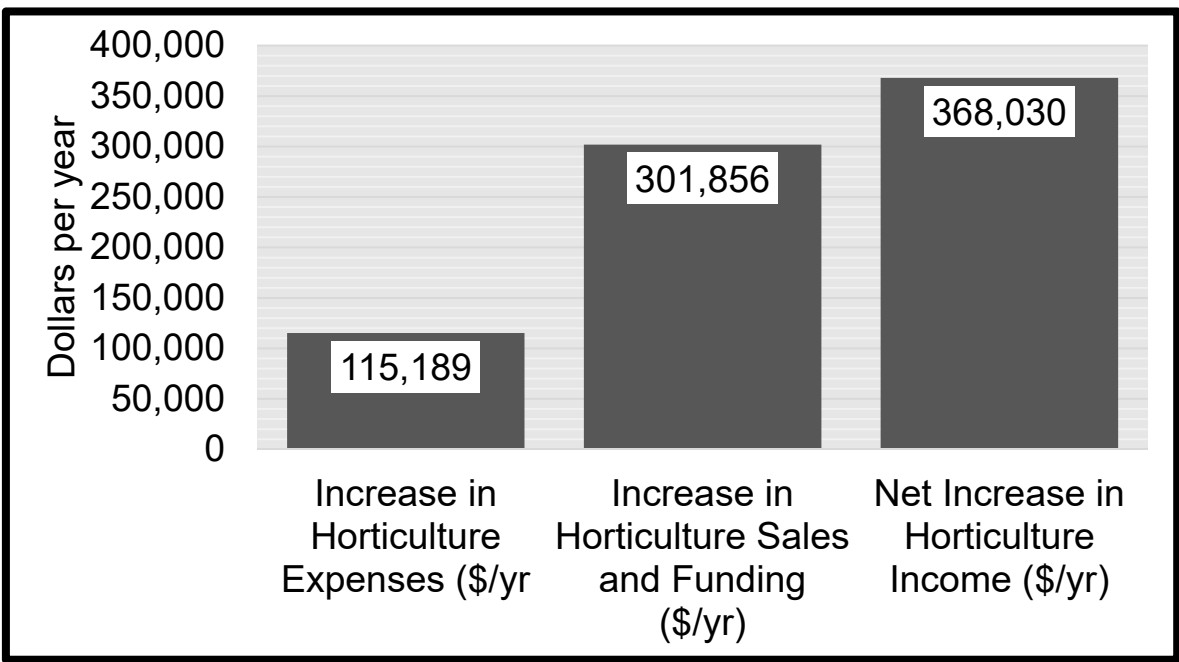

**Figure 1.** It shows the average annual increase in expenses, sales and funding, and income reported by adult attendees of horticulture events conducted at MSU-CREC from 2015 to 2019.

### 3.1. Changes in Horticulture Sales

The total increase in gross sales and project funding, as defined in Equation (2), averaged $USD 301,856/year (Figure 1). This value was used to create an IMPLAN scenario with increased annual sales in the greenhouse, nursery, and floriculture production sectors. The yearly increase in horticulture sales reported by participants at horticulture events conducted by MSU-CREC created total economic impacts of $USD 0.51 million in sales, five jobs, $USD 0.15 million income, and $USD 0.24 value-added (Table 1). Combined local, state, and federal taxes reached $USD 0.06 million.

**Table 1.** Economic impact summary of the annual increase in MSU-CREC participants' horticulture sales and project funding.

| Impact Type | Employment | Labor Income ($USDM) | Total Value Added ($USDM) | Output ($USDM) |
|---|---|---|---|---|
| Direct effect | 3 | 0.09 | 0.13 | 0.30 |
| Indirect effect | 1 | 0.04 | 0.06 | 0.12 |
| Induced effect | 1 | 0.02 | 0.05 | 0.09 |
| Total effect [1] | 5 | 0.15 | 0.24 | 0.51 |

[1] Total does not add up due to rounding.

### 3.2. Changes in Horticulture Expenses

Total horticultural expenses are the sum of floral registration fees and travel expenses to horticulture events conducted by MSU-CREC, averaging $USD 115,189/year (Figure 1). This horticulture expense created an IMPLAN scenario with higher expenses in the greenhouse, nursery, and floriculture production sectors. The annual increase in horticulture expenses conveyed by participants of horticulture events conducted by MSU-CREC generated total economic impacts of $USD 0.20 million in sales, two jobs, $USD 0.06 million income, and $USD 0.09 million value-added (Table 2). The combined local, state, and federal taxes were $USD 0.02 million.

**Table 2.** Economic impact summary of the annual increase in MSU-CREC participants' expenses on registration and travel to horticulture events.

| Impact Type | Employment | Labor Income ($USDM) | Total Value Added ($USDM) | Output ($USDM) |
|---|---|---|---|---|
| Direct effect | 1 | 0.04 | 0.05 | 0.12 |
| Indirect effect | 0 | 0.01 | 0.02 | 0.05 |
| Induced effect | 0 | 0.01 | 0.02 | 0.03 |
| Total effect [1] | 2 | 0.06 | 0.09 | 0.20 |

[1] Total does not add up due to rounding.

### 3.3. Changes in Horticulture Income

The sum of Equations (3) and (4) specify the net increase in savings and decreased costs, averaging $USD 368,030/year (Figure 1). An IMPLAN scenario with a net increase in savings was created in the greenhouse, nursery, and floriculture production sectors. The annual expansion in horticulture savings stated by participants of horticulture events conducted by MSU-CREC made a total economic impact of $USD 8.02 million in sales, 70 jobs, $USD 1.22 million in income, and $USD 2.04 million in value-added (Table 3). The combined local, state, and federal taxes reached $USD 0.32 million.

**Table 3.** Economic impact summary of the annual increase in MSU-CREC participants' horticulture savings and costs reduction.

| Impact Type | Employment | Labor Income ($USDM) | Total Value Added ($USDM) | Output ($USDM) |
|---|---|---|---|---|
| Direct effect | 47 | 0.37 | 0.65 | 5.07 |
| Indirect effect | 17 | 0.66 | 1.01 | 2.25 |
| Induced effect | 5 | 0.19 | 0.38 | 0.71 |
| Total effect [1] | 70 | 1.22 | 2.04 | 8.02 |

[1] Total does not add up due to rounding.

### 3.4. Combined Total Economic Impacts

The total economic impacts of the increase in annual horticulture sales, expenses, and incomes reported by participants of horticulture events conducted by MSU-CREC were compiled from the results shown in Tables 1–3. The combined economic impacts of the increase in annual gross sales, spending, and incomes of participants of horticulture programs at the MSU-CREC from 2015 to 2019 reached $USD 8.73 million/year (Table 4). The participation of horticulture professionals and producers in these horticulture programs also created 76 jobs/year. Other economic impacts created by horticulture participants included an income impact of $USD 1.43 million, a value-added impact of $USD 2.36 million, and a local, state, and federal tax impact of $USD 0.40 million.

**Table 4.** Combined total economic impacts summary of the annual increase in MSU-CREC participants' gross sales and project funding, expenses, and savings.

| Impact Type | Employment | Labor Income ($USDM) | Total Value Added ($USDM) | Output ($USDM) |
|---|---|---|---|---|
| Direct effect | 52 | 0.50 | 0.83 | 5.49 |
| Indirect effect | 19 | 0.71 | 1.08 | 2.42 |
| Induced effect | 6 | 0.22 | 0.45 | 0.83 |
| Total effect [1] | 76 | 1.43 | 2.36 | 8.73 |

[1] Total does not add up due to rounding.

### 3.5. Total Willingness to Pay

There were considerable estimates of the total willingness to pay (WTP) for the information learned from MSU-CREC horticulture programs. The total willingness to

pay amounted to \$USD 1.82 million/year. The assessments of the economic impacts of participants' increased horticultural sales, expenses, savings, and the total willingness to pay for information gained during horticulture events seem considerable compared to total funding for the horticulture programs.

### 3.6. Economic Impact of Public Spending

The combined state and federal funding for the horticulture programs at MSU-CREC averaged about \$USD 1.42 million/year. More than half comes from an annual federal appropriation for the conduct of horticultural research at MSU-CEC. The rest consists of state appropriations on the extension functions of the horticulture faculty at MSU-CREC.

The IMPLAN scenario capturing this annual spending generates additional economic impacts on the regional economy. The annual MSU-CREC public horticulture spending generated total economic impacts of \$USD 2.46 million in sales, 14 jobs, \$USD 0.70 million in income, and \$USD 1.06 million in value-added (Table 5). The combined local, state, and federal taxes were \$USD 0.20 million.

**Table 5.** Total economic impact summary of MSU-CREC public spending on horticulture programs.

| Impact Type | Employment | Labor Income ($USDM) | Total Value Added ($USDM) | Output ($USDM) |
|---|---|---|---|---|
| Direct Effect | 7 | 0.41 | 0.55 | 1.42 |
| Indirect Effect | 5 | 0.19 | 0.29 | 0.64 |
| Induced Effect | 3 | 0.11 | 0.22 | 0.40 |
| Total Effect [1] | 14 | 0.70 | 1.06 | 2.46 |

[1] Total does not add up due to rounding.

## 4. Discussion

We measured the economic impacts of horticulture programs from participants' responses at several horticulture events. The estimation procedures were organized by program. We estimated total changes in gross sales and project funding, expenses to register and travel, and an increase in saving and reduction in costs of participants of the horticulture research and extension programs at the Mississippi State University—Coastal Research and Extension Center for the past five years. The annual values were used in estimating the total economic impacts of private spending, sales, and incomes of participants in horticulture events.

Several assumptions were made during the estimation process. Changes in expenditures, sales, and incomes reported by participants in the ornamental horticulture field days and blueberry workshops were applied to other horticulture programs. These changes were also applied to the five years covered by this study.

Previous studies measured long-term econometric impacts of expenditures on research and extension on labor productivity in the agricultural sector in both developed and developing countries. The methodologies differed from the ones we used in our paper. It would have been desirable to compare our results with similar assessments conducted in other programs in the United States or other countries.

Estimates of the willingness to pay and economic impacts are indicators of the public perceptions of the economic contributions of the horticulture programs to the growth of the green industry in the region. The considerable willingness to pay for the information learned and the substantial economic impacts generated by the participation in horticulture events merit continued funding of the horticulture programs at the Mississippi State University-Coastal Research and Extension Center. In addition, the public spending on the horticulture program created other substantial economic impacts on the region.

It is suggested that similar evaluation methodologies be regularly conducted at horticulture events to further strengthen the measurement process and accuracy of the estimates. Objectives of the assessments for research and extension activities must be specific, measurable, attainable, realistic, and time-bound. The survey instrument must be comparable

for compilation and analysis. Another suggestion is putting together a few focus groups as another data gathering system.

**Author Contributions:** Conceptualization, B.C.P., P.R.K., C.E.H.C., E.K.B., E.T.S., J.D., G.B. and S.A.L.; methodology, B.C.P.; software, B.C.P.; validation, B.C.P.; formal analysis, B.C.P.; investigation, B.C.P., P.R.K., C.E.H.C., E.K.B., E.T.S., J.D., G.B. and S.A.L.; resources, B.C.P., P.R.K., C.E.H.C., E.K.B., E.T.S., J.D., G.B. and S.A.L.; data curation, B.C.P.; J.D., E.T.S., C.E.H.C.; writing—original draft preparation, B.C.P.; writing—B.C.P.; review and editing, B.C.P., P.R.K., C.E.H.C., E.K.B., E.T.S., J.D., G.B. and S.A.L.; visualization, B.C.P.; supervision, B.C.P.; project administration, B.C.P.; funding acquisition, P.R.K., C.E.H.C., E.K.B., E.T.S., J.D., G.B., B.C.P. All authors have read and agreed to the published version of the manuscript.

**Funding:** This research received no external funding.

**Institutional Review Board Statement:** Not applicable.

**Informed Consent Statement:** Not applicable.

**Data Availability Statement:** Not applicable.

**Acknowledgments:** This publication is a contribution of the Mississippi Agricultural and Forestry Experiment Station. This material is based upon work supported in part by the National Institute of Food and Agriculture, U.S. Department of Agriculture, Hatch project under accession number 232036.

**Conflicts of Interest:** The authors declare no conflict of interest.

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
