# Peer review of "Economic Impacts of Horticulture Research and Extension at MSU Coastal Research and Extension Center"

_horticulturae, doi:10.3390/horticulturae7080236_

Round 1

Reviewer 1 Report

There has been a lot of work done for this study to become en article. Many people has to been involved and the effort was quite high. It is somewhat interesting however in my opinion this is mostly localy (USA) important study. As a scholar from Europe I am not familiar with the subsidies system and financing of the so called by the author "extenstions" programs of any kinds. I don't know if asking of any comparison with European systems is appropriate, however maybe it is worth of another studies and new publications? 

Author Response

MINOR-SPELLING ERROR - The entire manuscript was reviewed by using a premium version of Grammarly. This software has its own dictionary. All relevant changes suggested were incorporated in the manuscript. 

MINOR-SPELLING ERROR - The built-in Editor program within Word also scanned the entire document. The spelling error was limited to "yr".  To correct this spelling error all of the "yr" was changed to "year". No more spelling error was found after that. 

RESEARCH PROGRAM FUNDING - The Mississippi Agricultural and Forestry Experiment Station (MAFES) is the agricultural research arm of Mississippi State University.  In 1887, the Hatch Act established the agricultural experiment station system, modeled on European stations, but with distinctly American interest in applied research. Later legislation provided for direct annual appropriations to each state to support its land-grant college.

EXTENSION PROGRAM FUNDING - The existence of land-grant colleges and experiment stations resulted in a growing logjam of knowledge that needed to be made available to the farmer and farm family in the field. A variety of activities including farmers' institutes, agricultural societies, and corn and tomato clubs were tried to meet these needs. In response, the Smith-Lever Act of 1914 established cooperative extension work. MSU Extension is the outreach arm of Mississippi State University.

Reviewer 2 Report

General Comments:

Interesting, important, and detailed manuscript about the economic impacts of five years of horticultural and extension activities of the MSU Coastal Research and Extension Center.

The manuscript is very straight forward with good approaches and informative results. There would have to be knit-picking in order to find any apparent flaws.

However, to improve the manuscript the authors could add sections in the methods and extend the section in the discussion dealing with uncertainties and factors that might skew their results, since the results are in-part based on surveys and self-reporting.

In addition, the reflection using the findings of other studies and their assessed economic impacts of horticultural and extension activities within the discussion would place the study in a relative context to other groups, universities, and stakeholders.

Author Response

MINOR-SPELLING ERROR - The entire manuscript was reviewed by using a premium version of Grammarly. This software has its own dictionary. All relevant changes suggested were incorporated in the manuscript.

MINOR-SPELLING ERROR - The built-in Editor program within word also scanned the entire document. The spelling error was limited to "yr". To correct this spelling error all of the "yr" was changed to "year". No more spelling error was found after that.

UNCERTAINTIES AND FACTORS THAT MIGHT SKEW THE RESULTS - Since data were mostly collected from voluntary surveys and self-reporting, large standard deviations from the means were observed.  The earlier analysis incorporated the standard deviations of each economic variable.  However, it was decided to drop the variations and use only the means of the available data in estimating economic impacts.  The reply to this comment is added to the manuscript in:
2. Materials and Methods
2.1 Sources of Primary Data.   

              Since data were primarily collected from voluntary surveys and self-reporting, large standard deviations from the means were observed.  The earlier analysis incorporated the standard deviations of each economic variable.  However, it was decided to drop the variations and use only the means of the available data to estimate economic impacts. 

FINDINGS OF OTHER STUDIES AND THEIR ASSESSED ECONOMIC IMPACTS OF HORTICULTURAL RESEARCH AND EXTENSION ACTIVITIES - Previous studies measured long-term econometric impacts of expenditures on research and extension on labor productivity in the agricultural sector in both developed and developing countries. The methodologies differed from the ones we used in our paper.  It would have been desirable to compare our results with similar assessments conducted in other programs in the United State or other countries. The reply to this comment is added to the manuscript in:

4. Discussion 

               Previous studies measured long-term econometric impacts of expenditures on research and extension on labor productivity in the agricultural sector in both developed and developing countries. The methodologies differed from the ones we used in our paper. It would have been desirable to compare our results with similar assessments conducted in other programs in the United States or other countries.

3.6. Economic Impact of Public Spending

              The combined state and federal funding for the horticulture programs at MSU-CREC averaged about $1.42 million/year. More than half comes from an annual federal appropriation for the conduct of horticultural research at MSU-CEC. The rest consists of state appropriations of the extension functions of the horticulture faculty at MSU-CREC.

Reviewer 3 Report

The concept and methodology are sound and clearly explained. The results are presented shortly and are easy to follow. I have no suggestions for further improvement. 

Author Response

Thank you for your prompt review of our manuscript.